# Vector Hysteresis Processes for Innovative Fe-Si Magnetic Powder Cores: Experiments and Neural Network Modeling

**Simone Quondam Antonio** [1,2,*] **, Francesco Riganti Fulginei** [3] **, Antonio Faba** [1,2] **, Francesco Chilosi** [2] **and Ermanno Cardelli** [1,2]

1    Department of Engineering, University of Perugia, Via G. Duranti 93, 06125 Perugia, Italy; antonio.faba@unipg.it (A.F.); ermanno.cardelli@unipg.it (E.C.)
2    CMIT Research Centre, Via G. Duranti 93, 06125 Perugia, Italy; f.chilosi@tamura-europe.co.uk
3    Engineering Department, Roma Tre University, Via Vito Volterra 62, 00146 Roma, Italy; francesco.rigantifulginei@uniroma3.it
*    Correspondence: simonequondam87@gmail.com or simone.quondamantonio@unipg.it; Tel.: +39-347-709-7272

**Abstract:** A thorough investigation of the 2-D hysteresis processes under arbitrary excitations was carried out for a specimen of innovative Fe-Si magnetic powder material. The vector experimental measurements were first performed via a single disk tester (SDT) apparatus under a controlled magnetic induction field, taking into account circular, elliptic, and scalar processes. The experimental data relative to the circular loops were utilized to identify a vector model of hysteresis based on feedforward neural networks (NNs), having as an input the magnetic induction vector B and as an output the magnetic field vector H. Then the model was validated by the simulation of the other experimental hysteresis processes. The comparison between calculated and measured loops evidenced the capability of the model in both the reconstruction of the magnetic field trajectory and the prediction of the power loss under various excitation waveforms. Finally, the computational efficiency of the model makes it suitable for future application in finite element analysis (FEA).

**Keywords:** magnetic powder core; vector hysteresis; magnetic measurements; neural network modeling; power loss prediction

## 1. Introduction

In recent years the development and the production of innovative technologies for soft magnetic cores has attracted a lot of scientific interest. On the one hand, iron-based alloys available in form of laminations still constitute a suitable solution for the manufacturing of electrical machines and magnetic components in the industrial and energy environment [1,2]. However, to meet the requirements of power consumption for high-frequency applications, and the need of either reducing the size and the weight of the cores or creating complex geometries, other technologies have been proposed and explored, such as ferrites [3–5], soft magnetic composites (SMC) [6], and more recently additive-manufactured cores [7,8] and magnetic powder cores [9–11]. In particular, the latter one is attracting a lot of interest since it matches good magnetic properties with high workability and suitable mechanical and thermal properties.

Innovative cores made with iron-silicon powder have been manufactured very recently and they are mostly registered brands. The main advantages of this technology are the linear behavior of the B-H characteristics in a wide range of excitation, the quite high value of the saturation induction (close to 1.5 T), the very narrow shape of the hysteresis loop, and the high resistivity. As a consequence, Fe-Si powder-based soft ferromagnets are mostly suitable for the production of magnetic components for power electronics such as buck/boost inductors and smoothing chokes for inverters, but also for various types of electrical machines for high-frequency applications. Indeed, the narrow hysteresis loops

together with high resistivity lead to a very high energy efficiency in a wide range of operating frequencies. Furthermore, as is highlighted in the paper, these materials are rather isotropic and can be effectively employed as the core of rotating machines. Finally, a minor application field is represented by wireless power transfer systems [12,13] and magnetic shielding components in the extremely low frequency (ELF) and super low frequency (SLF) bands, as an alternative to Ni-Fe alloys and electrical steel sheets [14].

A comprehensive characterization of the material properties and the availability of an accurate and efficient simulation tool are of great importance for the design and the development of electrical machines and magnetic components. Currently, the data provided by manufacturers are almost poor and only limited to scalar hysteresis properties. Indeed, in most of the cases, only the maximum value of the relative permeability, the saturation induction, and the maximum power loss at f = 50 Hz, eventually for few given induction levels, are declared. In addition, only sinusoidal supply signals are taken into account. However, it must be evidenced that in the real operating conditions, magnetic cores are not subjected to purely scalar excitation and the waveform of the supply signal in general is not sinusoidal. The characterization of the material behavior under rotational fields is not only important for rotating machines; indeed it is known that that type of excitation is also somehow found in transformers and reactors [15,16].

With the aim to give a contribution in this field of research, the present work describes the characterization of vector hysteresis processes under arbitrary excitation waveforms for a sample of innovative soft magnetic material based on Fe-Si powder, produced by Chang Sung Corporation® (CSC), Incheon, Korea. The material has not yet been settled on the commercial market and was provided only for experimental and scientific analysis purposes. A thorough experimental analysis was performed by means of a single disk tester (SDT) apparatus with the scope to show the material behavior under various types of vector excitations that may occur in practical operating conditions. For this reason, the measurements were performed under both circular and elliptic magnetic induction trajectories and with either sinusoidal or non-sinusoidal waveforms. The scalar hysteresis loops were also measured along two perpendicular directions, in this case with purely sinusoidal magnetic inductions. The experimental investigation can highlight the vector hysteresis properties of the material with a high degree of completeness. A detailed description of the experimental analysis, covering the characteristics of the sample and the testing equipment as well as the set of measurements recorded, is discussed in Section 2.

In addition to the experimental verification, a computationally efficient vector hysteresis model based on feedforward neural networks (NNs) was utilized to reproduce the behavior of the test sample and to predict power losses. The use of neural networks in simulating hysteresis phenomena has been extensively studied in the literature, mostly for scalar problems [17–23]. Indeed, NN-based approaches are computationally efficient simulation tools and can be easily formulated in either direct (H input–B output) or inverse (B input–H output) form. These features make them particularly adequate for matching with finite element analysis (FEA), thus allowing the simulation of magnetic devices in the time domain [24].

Some of the proposed approaches are actually coupled models, in which the total magnetization is given as the sum of a memoryless component (reversible) computed by the neural network, and an irreversible component determined by exploiting other hysteresis models, such as the Preisach model [17–19]. However, other authors used a standalone feedforward neural network to reproduce the hysteresis phenomenon with satisfactory results in one dimension (1-D) [20–22]. In most cases the memory storage mechanism is taken into consideration, exploiting feedback algorithms such that the previous value of the model output (or some other information about the "past history") is also given as an input. Other strategies to account for the memory storage property are based on the recurrent NNs, with increased model complexity, as reported in [23], but they are not considered in this work.

Despite the large amount of papers dealing with the reproduction of the scalar hysteresis phenomenon via NN-based models, the same approaches for vector hysteresis problems have been less explored [25]. The application of a neural system consisting of an assembly of feedforward networks was proposed in [26] to reproduce vector magnetization patterns for Fe-Si laminated steels, but the comparison with the experimental data only covered the rotational loops under circular magnetic induction with sinusoidal waveforms. In addition, the problem of power loss prediction has not been examined. The characterization of vector magnetization processes under elliptical B was recently studied in [27] for electrical steel, as well as the modeling of the material behavior via the NN approach, but again with purely sinusoidal waveforms.

The second scope of the work is therefore to deal with hysteresis modeling and power loss estimation, taking into account generic supply conditions. The vector NN model, similar to the one introduced in [26] and also discussed in [28], was implemented in a Matlab® computing environment and is presented in Section 3. It is seen that the model can be opportunely identified only by exploiting the rotational loops with circular magnetic induction trajectories, whereas the other measured processes can be used as test cases for validation.

The results of the model identification and the comparison between simulated and measured data are properly shown and discussed in Section 4.

## 2. Experimental Investigation

The material examined in this work is an innovative and performing Fe-Si magnetic powder alloy with distributed air gaps provided by Chang Sung Corporation® for scientific purposes. The main features declared by the manufacturer are very low power losses, especially for high-frequency applications, and a considerably wide region of linearity, i.e., the range of applied magnetic fields such that the material response can be considered linear. The aforementioned properties make the material adequate to meet the requirements found for several fields of usage, primarily the filtering inductors and the power reactors, the design of which via conventional laminated materials requires a careful design and calibration of controlled air gaps in the core. The material composition is 70% FSiCr and 30% CIP, where:

- FSC = iron 90% + silicon 8% + chromium 2% (by weight)
- CIP = carbonyl iron powder

The specimen under test was produced with a disk shape with a diameter of D = 68 mm and a thickness of d = 0.98 mm, and the density declared by the manufacturer was $\rho$ = 6300 kg/m$^3$. The final preparation of the sample, carried out in our laboratory, consisted of creating two perpendicular sensing coils for the measurements of the in-plane components of the magnetic induction vector, according to the Faraday–Neumann–Lenz law. The sensing coils, with three turns each, were fixed to the surface by applying a layer of transparent and insulating varnish.

The disk sample, shown in Figure 1a, was finally attached to a wood support and mounted on the single disk tester machine, the core of which is a stator of a single-phase induction motor with two perpendicular excitation windings, thus allowing the generation of arbitrary 2-D magnetic field excitations. The magnetic field probe consists of an array of three equally spaced bi-axial hall sensors, allowing the estimation of the field at the sample surface by compensation of the demagnetizing effect, similar to the one presented in [29].

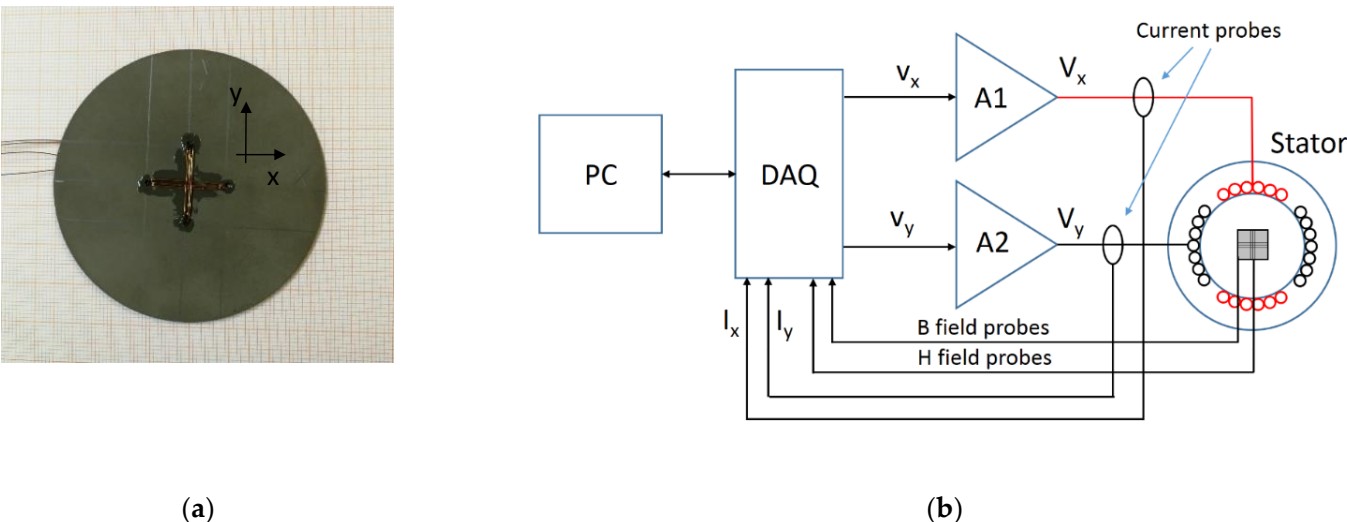

**(a)**　　　　　　　　　　　　　　　　　　　　　　　　　　**(b)**

**Figure 1.** Experimental measurements: (**a**) test sample after preparation and (**b**) block scheme of the testing apparatus: PC, DAQ HW, and power amplifiers A1 and A2 feeding the stator coils.

The two perpendicular windings of the stator are fed by two programmable linear 4-quadrant power amplifiers with frequency band DC—20 kHz and nominal voltage gain $A_v = 5$. A data acquisition board (DAQ), working in generation/acquisition mode was programmed in Matlab® to generate the input signals of the amplifiers and simultaneously detect the signals for the evaluation of H and B and for the control of the measurement system.

The complete list of the signals generated and measured, as well as the corresponding I/O analog ports configured on the DAQ hardware, are reported in Table 1. The two analog outputs (AO0, AO1) serve to transmit the applied voltage waveforms to the input of the power amplifiers. The maximum peak-to-peak voltage that can be generated by the DAQ is 20 V. The analog inputs AI0 and AI1 are connected to the sensing coils for the measurement of the magnetic induction components. The currents on the stator (excitation) coils are sensed at analog input ports AI2 and AI3 by means of two active current monitors, whereas the voltages applied to the stator are measured on ports AI4 and AI5 by means of two active differential voltage probes. Finally, the magnetic field components are measured by the three bi-axial hall sensors connected to analog input ports AI6 to AI20. The analog output (AO) ports are ground referenced single-ended connections, whereas all the analog input (AI) ports are configured as "floating," i.e., differential sense to maximize the signal-to-noise ratio. The block diagram of the SDT testing frame is shown in the Figure 1b, whereas the apparatuses used, with their corresponding models, are reported in the Table 2.

**Table 1.** Signals generated and measured by the DAQ in the single disk tester apparatus.

| Signals | Symbol | DAQ Ports |
|---|---|---|
| Input voltage to power amplifiers | $v_x(t)$, $v_y(t)$ | AO0, AO1 |
| Electro-motive force on B sensing coils | $e_{mx}(t)$, $e_{my}(t)$ | AI0, AI1 |
| Current on stator windings | $I_x(t)$, $I_y(t)$ | AI2, AI3 |
| Applied voltage on stator windings | $V_x(t)$, $V_y(t)$ | AI4, AI5 |
| Magnetic field sensor | $V_{Hx,i}(t)$, $V_{Hy,i}(t)$ i = 1, 2, 3. | From AI6 to AI20 |

**Table 2.** Instrumentation adopted in the SDT testing frame: apparatuses and corresponding models.

| Equipment | Model |
|---|---|
| Power amplifier | Kepco BOP 36-5 |
| Current probe | Rhode & Schwarz RT-ZC03 |
| Data acquisition module | NI USB 6363 BNC Type |
| Magnetic field sensor | Sentron 2SA-10 |
| Voltage probe | Micsig DP 10013 |

The core of the computer program that manages the generation/acquisition process is a digital feedback algorithm that allows the magnetic induction vector to be controlled. In particular, the operator defines the reference waveforms of the magnetic induction $B_{x,ref}(t)$ and $B_{y,ref}(t)$, the maximum percentage error between measured and reference signals, and the maximum number of iterations. The computer program automatically adjusts at each jth iteration of voltages $v_x^j(t)$ and $v_y^j(t)$ to be applied to the amplifiers in order to let the waveforms of the magnetic induction components converge towards reference ones. The voltage signals applied for the successive iteration, $v_x^{j+1}(t)$ and $v_y^{j+1}(t)$, are given from the ones applied in the previous iteration by adding the time-dependent incremental terms $\Delta_x^{j+1}(t)$ and $\Delta_y^{j+1}(t)$, which are proportional to both the difference between the reference and measured magnetic induction and the difference between their time derivatives:

$$\begin{cases} \Delta_x^{j+1}(t) = \dfrac{\max\left\{v_x^j(t)\right\}}{\max\left\{B_{x,ref}(t)\right\}} \cdot \xi \cdot \left(\alpha \cdot \Delta B_x + (1-\alpha) \cdot \Delta\dot{B}_x\right) \\[2mm] \Delta_y^{j+1}(t) = \dfrac{\max\left\{v_y^j(t)\right\}}{\max\left\{B_{y,ref}(t)\right\}} \cdot \xi \cdot \left(\alpha \cdot \Delta B_y + (1-\alpha) \cdot \Delta\dot{B}_y\right) \end{cases} \tag{1}$$

where $\alpha$ and $\xi$ are adjustable constants.

All the measurements reported in this work were performed at the frequency of 1 Hz and the input data were acquired at the sample rate of 360 samples/s. The frequency was set small enough that eventual dynamic effects were negligible and the loops could be considered quasi-static. The parameters used for the convergence of the feedback algorithm, optimized empirically during the calibration stage, were $\alpha = 0.97$ and $\xi = 0.75$, whereas the maximum deviation tolerated between the reference and measured waveforms of the magnetic induction was err_rel_max = 1.9%.

Four sets of experiments were performed, consisting of a family of hysteresis loops under given magnetic induction waveforms. Finally, for each set, the matrices of Hx, Hy, Bx, and By were recorded and the specific energy loss (energy per unit of mass) relative to each cycle was computed from the area enclosed by the Bx–Hx curve and By–Hy curve. The voltages applied to the stator coils, as well as the current absorbed by them, were only monitored throughout the measurement process.

- SET 1 is a family of rotational loops under circular magnetic induction obtained with sinusoidal Bx(t) and By(t) waveforms at the amplitudes of $B_0$ = 10 mT, 50 mT, 100 mT, 200 mT, 300 mT, 400 mT, 500 mT, and 600 mT.
- SET 2 is a set of three families of hysteresis loops under elliptic magnetic induction obtained with the following reference waveforms:

$$\begin{cases} B_{x,ref}(t) = B_0 \cos(2\pi f_0 t) \\ B_{x,ref}(t) = AR \cdot B_0 \cos(2\pi f_0 t) \end{cases} \tag{2}$$

Each family was obtained at the same values of $B_0$ as the ones applied in the previous set, with a constant value of the aspect ratio (AR). The three values of the aspect ratio were AR = 0.25, 0.50, and 0.75. Actually, SET 1 can be described by Equation (2) with AR = 1.

- SET 3 is a family of rotational loops measured with two-tone reference waveforms of the magnetic induction components, which consists of the sum of a fundamental tone at 1 Hz and a fifth-order harmonic, according to:

$$\begin{cases} B_{x,ref}(t) = B_0\cos(2\pi f_0 t) + m \cdot B_0\cos(10\pi f_0 t) \\ B_{x,ref}(t) = B_0\cos(2\pi f_0 t) + m \cdot B_0\cos(10\pi f_0 t) \end{cases} \tag{3}$$

where m = 0.15 is the modulation index.

The family of loops was determined for the same values of $B_0$, hence the peak value of the magnetic induction ranged from 11.5 mT to 690 mT.

- SET 4 consists of two families of scalar hysteresis loops, one measured along the x-axis and the other along the y-axis of the specimen. In the first case Bx(t) is sinusoidal and By(t) = 0, whereas in the second case By(t) is sinusoidal and Bx(t) = 0. The amplitudes considered of the sinusoidal waveform $B_0$ were equal to the ones from SET 1.

## 3. Numerical Modeling

The NN-based model of vector hysteresis presented here is a suitably trained neural system (NS) that computes the kth sample of magnetic field components Hx(k) and Hy(k) as a function of the kth sample of magnetic induction components Bx(k) and By(k). We chose the "inverse form" because it is more suitable for an eventual matching with Finite Element (FEM) solvers, according to the scalar potential formulation.

It turns out that a single feedforward neural network is not sufficient to reproduce magnetic hysteresis over a wide range of excitation with acceptable accuracy. For this reason, we proposed subdividing the computational domain (Bx–By plane) into regions with circular symmetry.

The domain of the magnetic induction vector is subdivided into a number of n regions with a circular crown shape defined by $B_{min}^j \le B < B_{max}^j$ with j = 1, ..., n, and a couple of two-input single-output feedforward neural networks with one hidden layer with N artificial neurons are assigned to each circular crown, one for the computation of the x component and the other for the y component of the magnetic field, as a function of both Bx(k) and By(k). The jth couple of sub-networks is enabled for the computation of the magnetic field vector only if the magnetic induction vector belongs to the jth circular crown. The shape of the regions is suggested by the types of excitations used to train the neural networks: the quasi-static rotational loops with circular vector B.

The working principle of the NS can be schematized according to the block diagram shown in Figure 2.

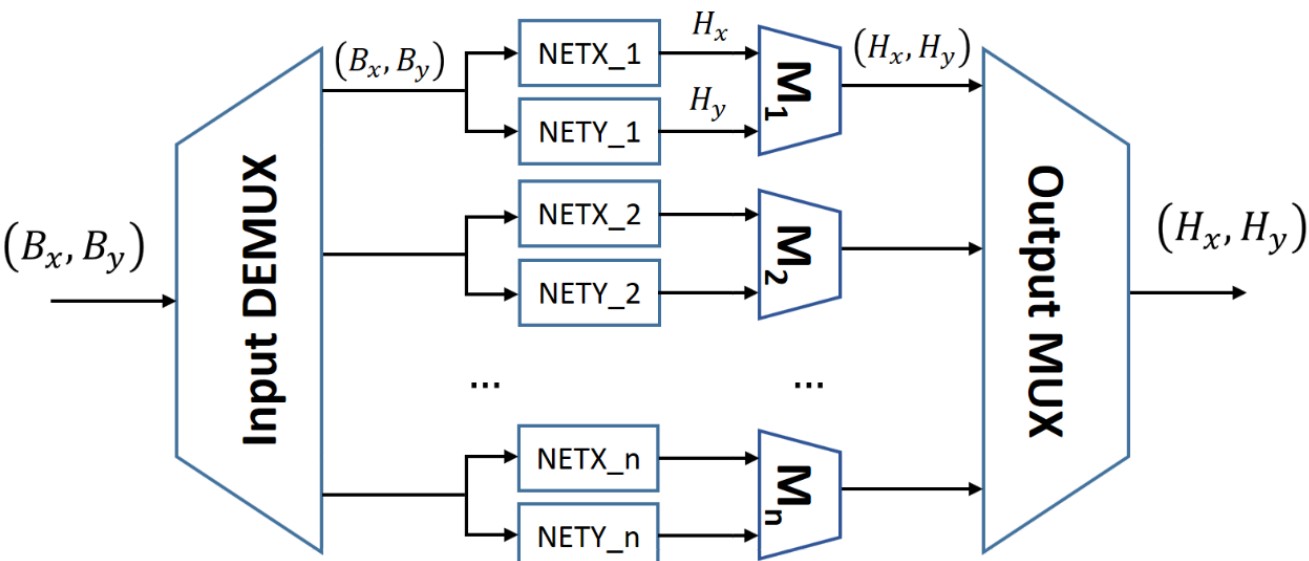

**Figure 2.** Block diagram of the theoretical neural system.

An input de-multiplexer assigns the magnetic induction vector (Bx, By) to the input of both feedforward neural networks pertaining to the proper circular crown. The components of the magnetic field, computed by the couple of NNs, are collected by the multiplexer Mj and finally sent to the output. The vector model was inspired by the one proposed in [23], however, it differs in terms of the architecture and formulation of the single sub-networks.

The most critical issues in reproducing hysteresis phenomena via neural network approaches are mostly the numerical stability and the robustness, which are related in a complex manner to the characteristics of the input, the formulation of the model and the feedforward neural networks, and the training procedure. Hence, the number of circular crowns, their amplitudes, and the number of neurons of the scalar NNs were carefully optimized with the aim of minimizing the model complexity and the computational resources to be allocated. The only magnetization processes involved in the definition of the model architecture and in the training procedure are the rotational loops measured under circular magnetic inductions of SET1, described in the previous paragraph. The following results were obtained:

- Number of circular crowns: n = 4.
- Amplitudes: $0 \leq B < 90$ mT; $90 \leq B < 290$ mT; $290 \leq B < 490$ mT; $B \geq 490$ mT.
- Number of neurons for each scalar net: N = 20.

The NS was identified by only eight patterns in a two-by-two subdivision in each circular crown region. The trajectories of the magnetic induction field, applied to the input of the NS, are displayed in Figure 3.

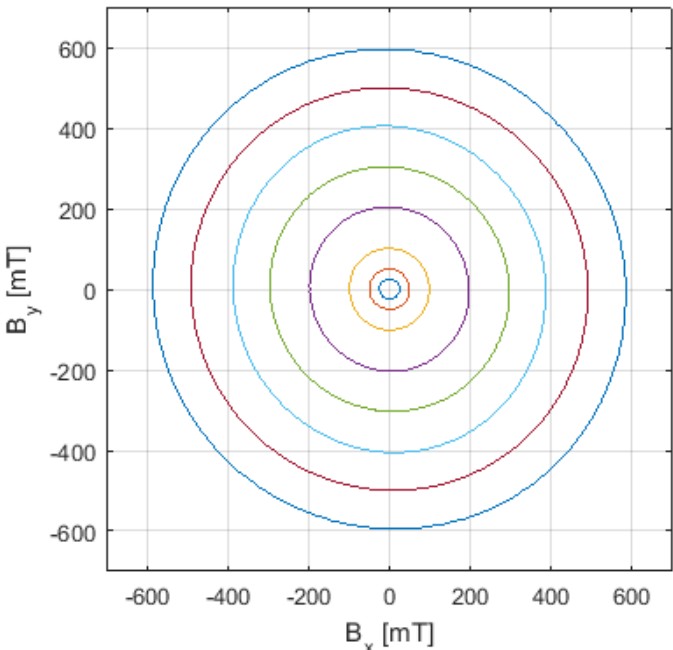

**Figure 3.** Circular trajectories of the magnetic induction vector measured for SET 1 utilized to train the NS.

The generic jth couple of sub-feedforward NNs (NETX_j, NETY_j) that can be viewed as a 2-D vector network NET_j were trained with the rotational loop data belonging to the jth circular crown plus the adjacent first neighbors. The loops of SET 1 involved in the training of each one of the four vector networks are listed properly in Table 3. The subsets were partially overlapping each other to stabilize the response of the networks near the boundary between regions.

**Table 3.** Loops for NS identification subdivided into partially overlapping sets for the training of single couples of NNs.

| Loop ID | 1 | 2 | 3 | 4 | 5 | 6 | 7 | 8 |
|---|---|---|---|---|---|---|---|---|
| **Amplitude** | 10 mT | 50 mT | 100 mT | 200 mT | 300 mT | 400 mT | 500 mT | 600 mT |
| **NET_1** | x | x | x | | | | | |
| **NET_2** | | x | x | x | x | | | |
| **NET_3** | | | | x | x | x | x | |
| **NET_4** | | | | | | x | x | x |

Let us conclusively specify that the training of each couple of scalar feedforward neural networks was performed with 15,000 epochs using the Levemberg–Marquardt algorithm. The number of samples used to train a single feedforward neural network is $2n_{loop}SPP$, where $SPP = 360$ (samples per period) and $n_{loop}$ is the number of loops that is either equal to 3 for the couple of NNs of NET_1 and NET4, or equal to 4 for the couple of NNs of NET_2 and NET_3, according to Table 3.

The accuracy of the identification stage was validated by the comparison of the computed trajectories of the H field with the measured ones. This comparison is illustrated in Figure 4 for some of the rotational loops of SET 1.

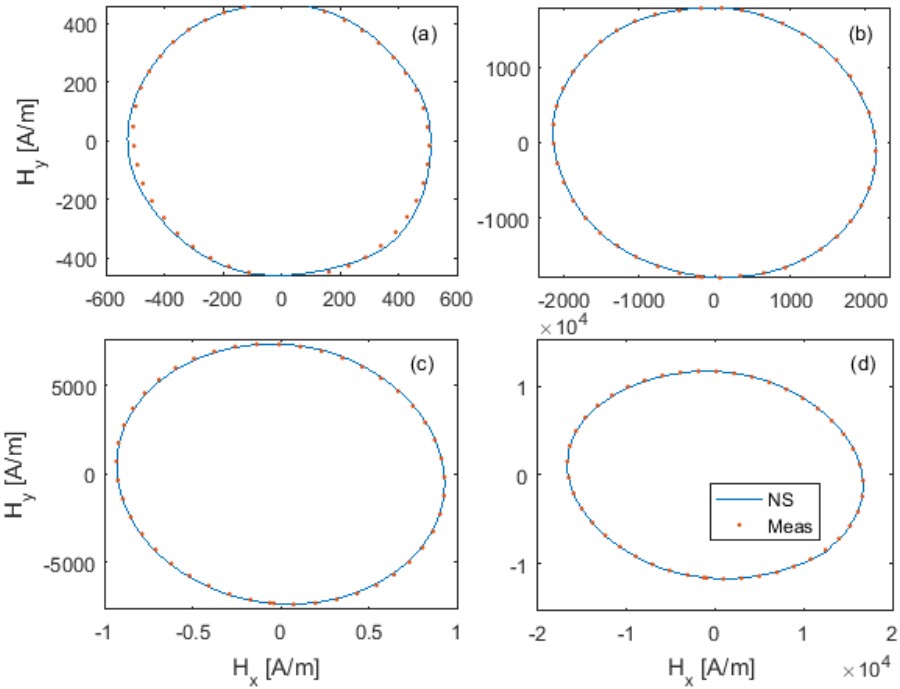

**Figure 4.** Comparison between the computed (blue solid line) and the measured (red dots) trajectories of H relative to the rotational loops of SET 1, where B = 10 mT (**a**), B = 100 mT (**b**), B = 400 mT (**c**), and B = 600 mT (**d**).

It can be observed that the trajectories of the magnetic field were not precisely circular and therefore the response of the material was not exactly isotropic. Indeed, if we consider the major loop, measured for B = 600 mT with a maximum error of 1.9%, the modulus of the magnetic field vector varies from a maximum value of 16.66 kA/m and a minimum value of 11.68 kA/m. The maximum value, found close to the x axis, identifies the hard magnetization direction, whereas the minimum value, found close to the y axis, identifies the easy magnetization direction. Since the NS was trained to reproduce the distorted H-field trajectories produced by the circular B-field trajectories, the effects of the magnetic anisotropy should have been correctly modeled.

## 4. Discussion

The NS, opportunely trained via the rotational loops, was validated by the reproduction of the magnetic field trajectories and the prediction of energy losses, under various types of supply conditions. In the test cases described hereafter, the sequences Bx(k) and By(k) applied to the input of the NS were directly taken from the measurements.

### 4.1. Elliptic Magnetic Induction

The first test case consisted of the simulation of the families of concentric elliptic loops for each one of the considered values of the aspect ratio. The measured trajectories of the magnetic induction, applied as input to the NS, are displayed in Figure 5. On the other hand, the measured and the simulated trajectories of the magnetic field vector on the Hx–Hy plane relative to the elliptic loops with AR = 0.25 are represented in Figure 6 for some values of the corresponding amplitude of Bx (major axis of the ellipse), covering the entire range examined. The same kind of magnetic field curves but relative to the other values of the aspect ratio are illustrated in Figures 7 and 8.

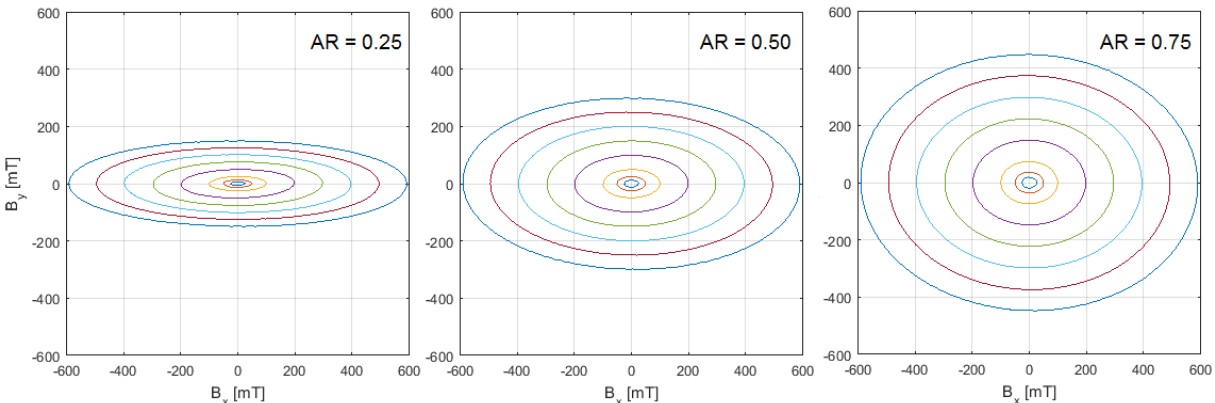

**Figure 5.** Families of elliptic loops relative to SET 2 measured at different amplitudes for each one of the constant values of the aspect ratio AR = 0.25 (**left panel**), AR = 0.50 (**center panel**), and AR = 0.75 (**right panel**).

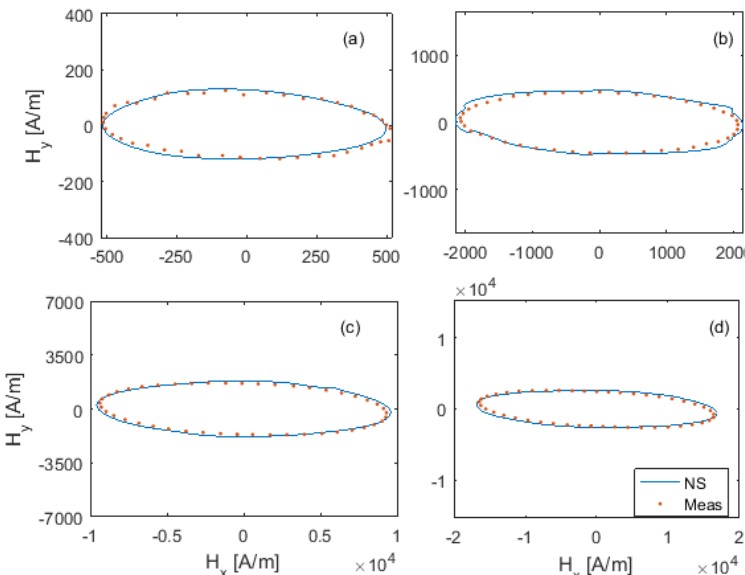

**Figure 6.** Comparison between simulated and measured trajectories of the H field relative to the elliptic loops with AR = 0.25 for the following values of the major half-axis: Bx = 10 mT (**a**), Bx = 100 mT (**b**), Bx = 400 mT (**c**), and Bx = 600 mT (**d**).

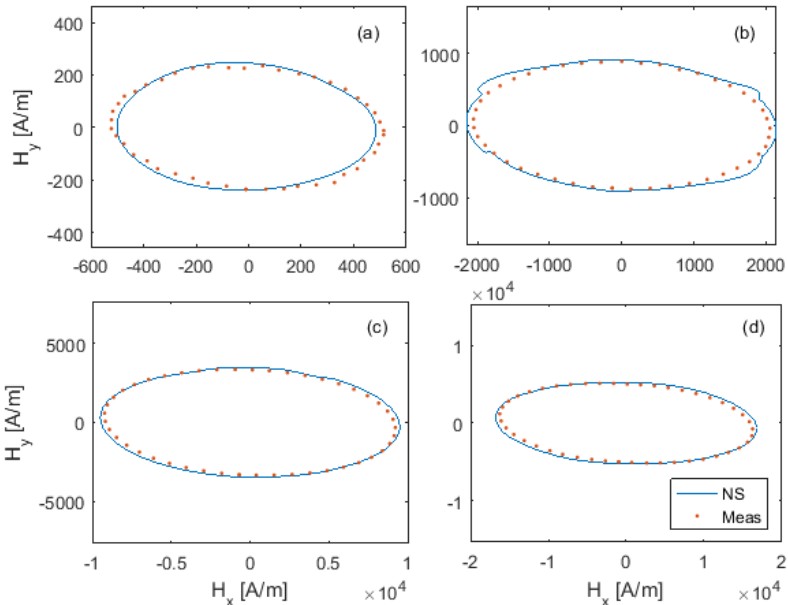

**Figure 7.** Comparison between simulated and measured trajectories of the H field relative to the elliptic loops, where AR = 0.50 for the following values of the major half-axis: Bx = 10 mT (**a**), Bx = 100 mT (**b**), Bx = 400 mT (**c**), and Bx = 600 mT (**d**).

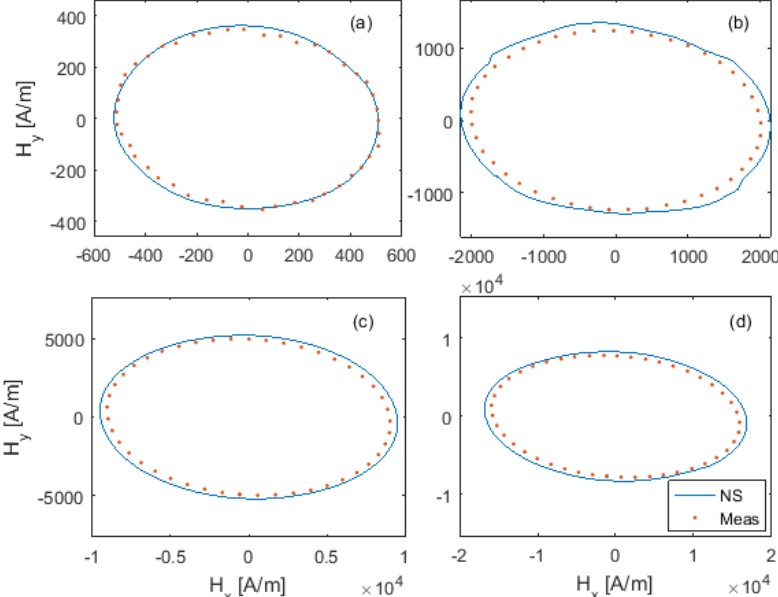

**Figure 8.** Comparison between simulated and measured trajectories of the H field relative to the elliptic loops, where AR = 0.75 for the following values of the major half-axis: Bx = 10 mT (**a**), Bx = 100 mT (**b**), Bx = 400 mT (**c**), and Bx = 600 mT (**d**).

The reconstructed curves were rather close to the measured ones for all the values of the aspect ratio examined. It must be noticed that when the input de-mux changes the couple of scalar feedforward neural networks for the calculation of Hx(k) and Hy(k) throughout the magnetization process, some discontinuities of the derivative to the curve traced by the H-field vector may occur. With the aim of reducing the probability of having such discontinuities, the rotational loops adjacent to the circumference that separated regions j and j + 1 were involved in the identification of both NET_j and NET_j+1 (where j = 1, 2, 3). For this reason, the computational flaw derived from the sample-by-sample selection of the active NNs was found only for Bx = 100 mT and did not significantly affect

either the accuracy in the magnetic field reconstruction or the computational stability of the model.

In order to make the model suitable for future implementation in FEM-based solvers, the prediction of the energy loss was also an important issue that deserved to be investigated comprehensively. For this reason, the energy loss per unit of mass as a function of the amplitude of Bx was determined from both the measured and the simulated loops. The comparison is displayed in Figure 9 for each value of AR.

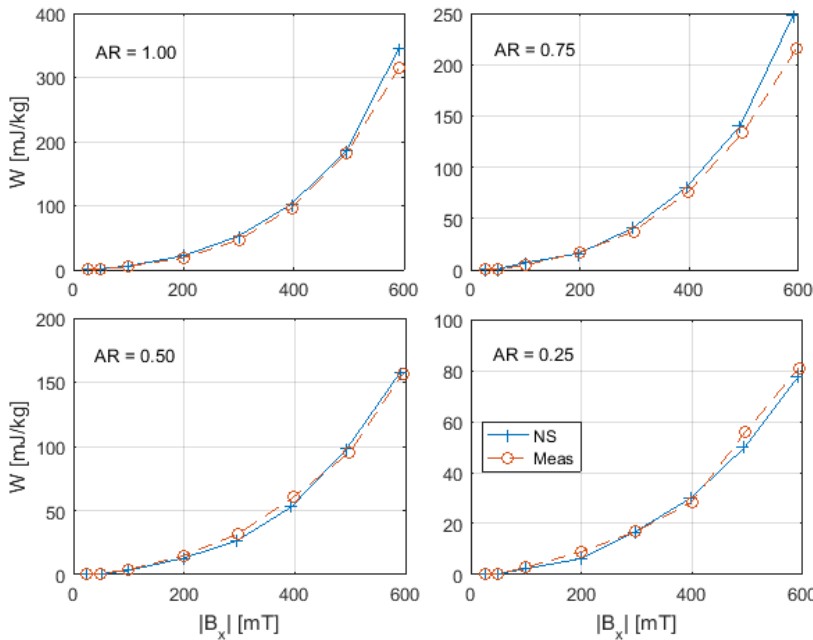

**Figure 9.** Comparison of the simulated and measured energy-loss curves for different value of the aspect ratio.

The case AR = 1 is relative to the rotational loops and the agreement between the simulated and the measured curves reflects mostly the accuracy of the training procedure. However, it must be specified that the loop area, and consequently the energy loss, was not involved in any way during the training of the model. The loss prediction for the other values of AR is even more interesting, since the field trajectories under elliptic B were not taken into account for NS identification. As one can view in Figure 9, the computed loss curves were very close to the measured ones. We computed the error as the absolute difference between the measured and simulated values of the energy loss. The absolute deviation averaged over all the curves was 7.9 mJ. The maximum deviation was found for AR = 0.75 and Bx = 600 mT, where the NN model tended to overestimate the energy loss by 29 mJ, corresponding to 13% of the measured value.

Finally, it is interesting to point out that the relationship between the energy lost and the aspect ratio was almost linear if the fixed level of the magnetic induction was higher than 300 mT. Indeed, the progression of the measured energy lost at the induction of 600 mT, here indicated with $W_{600}^m$, for AR = 0.25, 0.5, and 1 was 81.0 mJ/kg, 156.9 mJ/kg, and 315.2 mJ/kg, respectively. Similarly, the progression of the simulated energy lost at 600 mT, indicated similarly with $W_{600}^m$ for the same values of the aspect ratio, was 78.0 mJ/kg, 157.0 mJ/kg, and 344 mJ/kg, respectively. Similar values can be found if one considers other levels of Bx higher than 300 mT. For lower levels of induction, the behavior of the energy loss versus the aspect ratio was less regular, because the shape of the curves W(Bx) slightly changed with AR.

### 4.2. Two-Tone Magnetic Induction

The second test case investigated dealt with the analysis of vector hysteresis processes obtained with a distorted magnetic induction, which was characterized by a fifth-order harmonic added to the fundamental tone. The experimental characterization under non-sinusoidal supply signals aimed to show the material response emulating the working conditions that are found in several application fields, such as inverter-driven devices and filtering inductors. In addition, the capability of the NS to reproduce the material behavior under distorted excitations was also checked by comparison with measured data. This validation example was important for verifying and stressing the computational stability, the accuracy, and the robustness of the model.

The trajectories of the magnetic induction vector relative to the concentric loops of SET 3 are displayed in Figure 10, whereas the comparison between the simulated and measured trajectories of the magnetic field is shown in Figure 11.

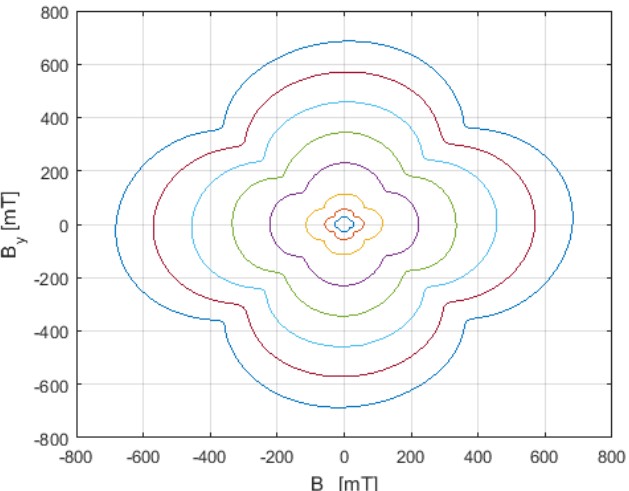

**Figure 10.** Family of rotational loops measured under two-tone magnetic induction waveforms (SET 3).

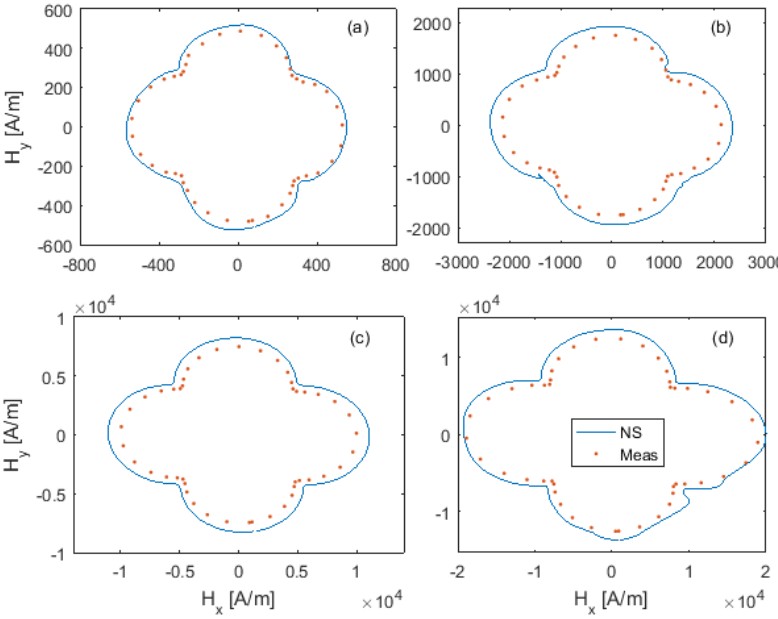

**Figure 11.** Comparison between simulated and measured trajectories of the H field relative to the rotational loops under two-tone magnetic inductions for different values of the amplitudes of the fundamental component B0 = 10 mT (**a**), B0 = 100 mT (**b**), B0 = 400 mT (**c**), and B0 = 600 mT (**d**).

The H-field curves simulated by the NS model were quite close to the measured ones in the entire range of excitations examined, but the modulus of the computed magnetic field was most of the time slightly higher than the measured one. Furthermore, the computational flaw observed in the previous test case seemed to no longer occur.

The specific energy loss was evaluated as a function of the peak value of the magnetic induction modulus from both the measured and the simulated data. The comparison between the curves computed from the simulated loops and those computed from the experimental ones is illustrated in Figure 12.

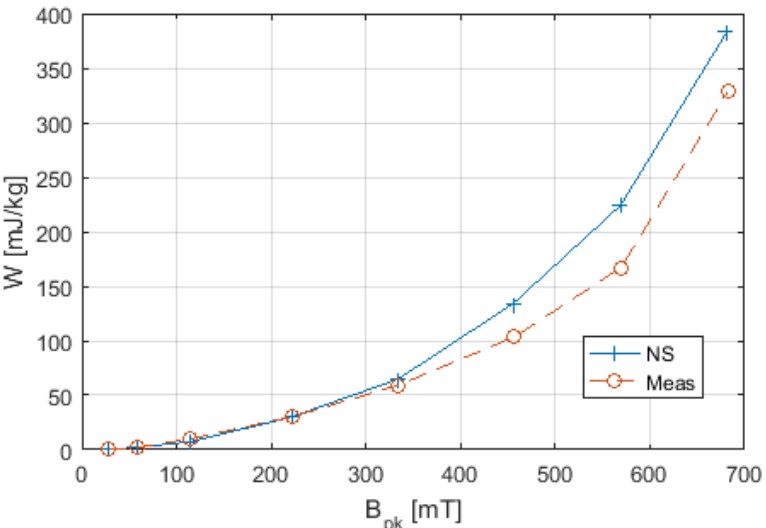

**Figure 12.** Comparison of the simulated and measured energy-loss curves relative to the two-tone magnetic inductions (SET 3).

The energy losses predicted by the NS model were very close to the ones measured up to $B_{pk}$ = 345 mT. Above that value the NS tended to overestimate the measured losses and the maximum percentage deviation found for $B_{pk}$ = 575 mT was 22%.

### 4.3. Scalar Magnetic Induction

As final step of our investigation, scalar hysteresis processes were also simulated by the NS. Since the physical phenomena that take place are actually different respective to those involved in rotational hysteresis mechanisms, the accurate reproduction of the scalar loops needed to be modeled separately, for instance by using a dedicated sub-routine that needs a separate identification procedure. However, it is shown in the following that some interesting features of the scalar loops, such as the first magnetization curve and the relative permeability, could be obtained by the standalone neural system. We simulated the scalar hysteresis loops along the x- and the y-axes setting the sequences of the magnetic induction Bx(t) and By(t) recorded for SET 4 as input to the NS separately.

From the measured and the simulated data we first determined the first B–H magnetization curves along both the 0° and the 90° directions, connecting the vertices of the hysteresis loops along the x- and the y-axes, respectively. The first magnetization curves are shown in Figure 13a, where the slight anisotropy character of the material, already evidenced by the analysis of the rotational loops, was again confirmed. Indeed, the slope of the first magnetization curve measured along the y-axis (90°) was sensibly higher than the slope of the fist magnetization curve measured along the x-axis (0°). The data obtained by the NS simulations correctly predicted the first magnetization curve without any additional identification procedures.

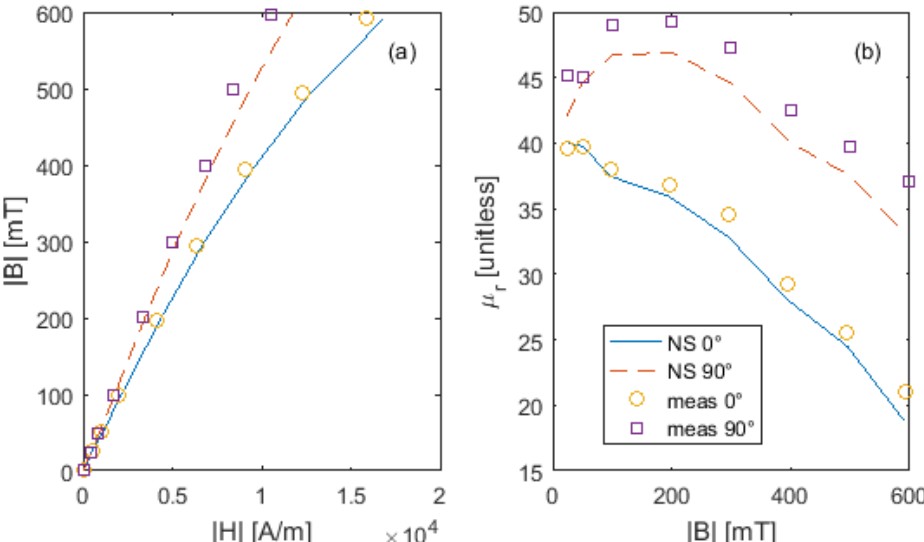

**Figure 13.** (**a**) First magnetization curve along both the x- and the y-axis computed from either the simulated or the measured scalar hysteresis cycles. (**b**) Relative permeability versus the amplitude of the magnetic induction evaluated along both the x- and the y-axis from either the simulated and the measured scalar hysteresis cycles.

The relative permeability was then calculated as the slope of the first magnetization curve divided by $\mu_0$ from either the measured or the simulated data along both the x and the y directions. The results are displayed in panel (b) of the same figure, where $\mu_r$ was plotted versus the amplitude of the magnetic induction. The magnetic anisotropy of the material was confirmed by the analysis of the relative permeability, which, along the y-axis, was up to 1.8 times higher the values found for the x-axis. In addition, the maximum value of the relative permeability along the y-axis was found for $|B| = 200$ mT, whereas along the x-axis the maximum was found for $|B| = 50$ mT. The substantial agreement between measured and simulated data was again confirmed.

It has to be specified that purely scalar excitations cannot be thoroughly represented by the NS, which would always produce an output magnetic field collinear with the input magnetic induction, and this is not exactly true in practice. For this reason, other features of the purely scalar hysteresis processes, such as the remanence, the coercivity, and the energy losses, cannot be conveniently obtained by the NS alone. Nevertheless, it must be said that in practice purely scalar excitations very rarely occur, and that in most of the cases magnetic induction trajectories are elliptical, eventually with an aspect ratio with a small value. These kinds of processes present a level of energy loss that is very close to scalar ones and can be effectively simulated by the NS, as discussed in Section 4.1.

Let us finally point out that all the simulations presented in the paper were performed on the same PC equipped with a CPU Intel[®] Core™ i7—2670 QM @ 2.20 GHz, with 8 GB of RAM memory and 64 bit operating system. On this machine the computation of the couple of output values, corresponding to the nth samples of Hx and Hy, requires about 14 milliseconds, and the total sample rate of the NS is 143 samples/s. The computational efficiency could be further improved by programming the model at a lower level of abstraction, and this is suggested for future implementation of the model in finite element solvers.

## 5. Conclusions

We presented a thorough vector experimental characterization for an innovative soft ferromagnetic material based on Fe-Si magnetic powder and distributed air gaps, as well as a dedicated technique to reproduce measured magnetization processes via a neural network approach.

The proposed vector hysteresis model was identified only using a family of eight quasi-static rotational loops with circular magnetic inductions obtained by the experiments. It turns out that the minimal training set adopted was sufficient to allow the reproduction of the elliptic loops for different aspect ratios and other arbitrary excitations with satisfactory accuracy. The energy losses were also correctly predicted in most of the test cases examined. For highly distorted excitation waveforms the energy losses simulated by the neural network-based model were a little higher than the measured ones and shall be considered as an upper limit.

The simulation of the scalar processes, performed by the NS without any additional identification procedure, led to the characterization of only partial features of the hysteresis loops, such as the relative permeability function and the first magnetization curve. Nevertheless, the behavior of the material could be conveniently emulated by applying elliptical loops with a very small aspect ratio, as usually happens in practice, allowing the thorough computation of the field trajectories and power losses with high accuracy.

Although the material examined in this work was only weakly anisotropic, we are confident that the proposed modeling technique, thanks to its flexibility and scalability, can be extended to other materials with higher magnetic anisotropy or to cover a wider range of excitations. Indeed, if more complex anisotropy patterns have to be modeled, larger feedforward neural networks with more neurons, or more hidden layers, or both can be adopted. In addition, the number of circular crowns can be increased to further partialize the input domain.

**Author Contributions:** Conceptualization, F.R.F. and S.Q.A.; methodology, F.R.F.; software, S.Q.A.; validation, S.Q.A.; investigation, S.Q.A. and F.R.F.; resources, E.C. and F.C.; data curation, S.Q.A.; writing—original draft preparation, S.Q.A.; writing—review and editing, E.C., A.F., and F.R.F.; supervision, E.C.; funding acquisition, E.C. and A.F. All authors have read and agreed to the published version of the manuscript.

**Funding:** This research was founded by means of internal financial resources of the CMIT Research Centre, and jointly co-founded by the University of Perugia—Engineering Department and the Tamura Corporation.

**Institutional Review Board Statement:** Not applicable.

**Informed Consent Statement:** Not applicable.

**Acknowledgments:** We acknowledge the Chang Sung Corporation®, from which we received the specimen of ferromagnetic material examined in the paper, suitably prepared for the experimental investigation.

**Conflicts of Interest:** The authors declare no conflict of interest.

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
