# Peer review of "Vector Hysteresis Processes for Innovative Fe-Si Magnetic Powder Cores: Experiments and Neural Network Modeling"

_magnetochemistry, doi:10.3390/magnetochemistry7020018_

Round 1

Reviewer 1 Report

  The authors present “Vector Hysteresis Processes for Innovative Fe-Si Magnetic Powder Cores: Experiments and Neural Network Modelling”, which systematically described the experimental measurements, numerical modelling, elliptic magnetic induction, two-tone magnetic induction and scalar magnetic induction. The manuscript itself is interesting, scientific and useful investigation for the power electronics, such as buck/boost inductors, smoothing chokes for inverters, but also various types of electrical machines for high frequency applications. However, before this manuscript is accepted there are a lot of problems need to be addressed:

  1. The English needs to be further polished in the revised manuscript (such as: panelï¼› 1ï¼›figureï¼›in section IVï¼›.…. etc).
  2. The content presented in Lines 5-8 is incorrect. In addition, the format of references in the manuscript content is as [7], [8] must be revised to [7,8].
  3. Table 1 is not described in the text of the manuscript. In addition, the relationship and links between Table 1 and Figure 1(b) need to be explained in more detail.
  4. The authors need to further explain why only Bx = 100 mT in Figures 6-8 has a computational flaw?
  5. The authors need further explanation:The average percentage error found is below 5%, whilst the maximum error, that occurs for AR = 0.75 and Bx = 600 mT is 9.6%?
  6. The authors need further explanation:Let us consider the progression of W600, i.e. the measured energy lost corresponding to Bx = 600 mT, as the aspect ratio doubles. For AR = 0.25, 0.5, 1 it has been found W600 = 81.0 mJ/kg, 156.9 mJ/kg, 315.2 mJ/kg, then the energy also doubles.
  7. (1) In the content of the manuscript, the authors need to explain NS 0°, NS 90°, meas 0°, meas 90° in Figure 13(b)?(2) The authors need to confirm and further analyze the results of NS and meas in Figure 13(b).

  I recommend that the paper will be published in magnetochemistry with minor revision.

Author Response

Dear reviewer, we have appreciated your help for improving the technical quality of our manuscript and we thank you for the useful questions and comments. In the following we list the answers given point-by-point. All the modifications made to the original manuscript have been written in red text.

  • The English needs to be further polished in the revised manuscript (such as: panelï¼› 1ï¼›figureï¼›in section IVï¼›.…. etc).

RESPONSE: We have read the manuscript and polished the English. The modifications made throughout the text have been written in red colour.

  • The content presented in Lines 5-8 is incorrect. In addition, the format of references in the manuscript content is as [7], [8] must be revised to [7,8].

RESPONSE: We have modified the author’s names according to the format. We have also marked the corresponding author as specified in the template and we have modified the indication of the references, as suggested.

  • Table 1 is not described in the text of the manuscript. In addition, the relationship and links between Table 1 and Figure 1(b) need to be explained in more detail.

RESPONSE: Table 1 is now described in the revised manuscript (from row 142 to row 154) helping also the relationship between the components listed in table 2 and the figure 1(b). To this end, the caption of Figure 1 (b) has been also extended.

  • The authors need to further explain why only Bx = 100 mT in Figures 6-8 has a computational flaw?

RESPONSE: As it has been now evidenced in the revised manuscript (from row 313 to row 321), the computational flaw appears when the input magnetic induction vector crosses the boundary between the first and the second circular crown region. The discontinuities are in any case very small because the rotational loops adjacent to the boundary have been involved in the training of the NNs belonging to both the regions.

  • The authors need further explanation:The average percentage error found is below 5%, whilst the maximum error, that occurs for AR = 0.75 and Bx = 600 mT is 9.6% ?

RESPONSE: We have corrected the value and further commented the results shown in figure 9 (from row 339 to row 344).

  • The authors need further explanation:Let us consider the progression of W600, i.e. the measured energy lost corresponding to Bx = 600 mT, as the aspect ratio doubles. For AR = 0.25, 0.5, 1 it has been found W600 = 81.0 mJ/kg, 156.9 mJ/kg, 315.2 mJ/kg, then the energy also doubles.

RESPONSE: In the revised manuscript we have better explained this concept (from row 345 to row 352). We have pointed out that if the magnetic induction is sufficiently high (above 300 mT) the relationship between the energy lost and the aspect ratio is almost linear.

  • (1) In the content of the manuscript, the authors need to explain NS 0°, NS 90°, meas 0°, meas 90° in Figure 13(b)?(2) The authors need to confirm and further analyze the results of NS and meas in Figure 13(b).

RESPONSE: (1) We have better explained the results shown in figure 13 (b), where we have described the differences found along the two main in-plane directions in terms of both the measured and simulated data. (2) We have pointed out that the results shown are a synthesis of the results obtained comparing the simulated and measured hysteresis loops along the two directions. In addition, we have better commented the relationship between the first magnetization curve shown in panel (a) and the relative permeability shown in panel (b). The modifications required can be found from row 400 to row 415 of the revised manuscript.

Reviewer 2 Report

It would be useful to appear clearly the number of samples that was used for the training of NN. If it is 2 for each circular crown this number possibly is very small. The reason that is used separate NN for each circular crown would also be useful to appear clearly. The values of parameters for the algorithm convergence would be useful to be noted.

Author Response

Dear reviewer, we have appreciated your help for improving the technical quality of our manuscript and we thank you for the useful questions and comments. In the following we list the answers given point-by-point. All the modifications made to the original manuscript have been written in red text.

  • It would be useful to appear clearly the number of samples that was used for the training of NN. If it is 2 for each circular crown this number possibly is very small.

RESPONSE: A more detailed description of the training procedure has been included in the revised paper, according to this comment of the Reviewer (from row 264 to row 267). Actually, the number of samples used to train each neural network are 2*SPP * nLoops (where SPP = 360 are the samples per period and nLoops = the number of loops). We have nL = 3 for the nets: NET_1 and NET_4, while nL = 4 for the networks NET_2 and NET_3, according to table 3.

  • The reason that is used separate NN for each circular crown would also be useful to appear clearly.

RESPONSE: in the revised manuscript we have clearly specified the reason why it is necessary to subdivide the computational domain into regions. Indeed, it can be proven empirically that a single couple of NNs cannot reproduce the hysteresis phenomenon over a wide range of excitations with acceptable accuracy. The modified text can be found from row 217 to row 228.

  • The values of parameters for the algorithm convergence would be useful to be noted.

RESPONSE: we have better indicated the meaning and the values used for the parameters of the feedback algorithm employed in the control of the measurement system (from row 176 to row 181).